# Sequence variants affecting the genome-wide rate of germline microsatellite mutations

Snaedis Kristmundsdottir [1,2], Hakon Jonsson [1], Marteinn T. Hardarson [1,2], Gunnar Palsson [1], Doruk Beyter[1], Hannes P. Eggertsson [1], Arnaldur Gylfason [1], Gardar Sveinbjornsson [1], Guillaume Holley[1], Olafur A. Stefansson[1], Gisli H. Halldorsson [1,3], Sigurgeir Olafsson [1], Gudny. A. Arnadottir [1,4], Pall I. Olason[1], Ogmundur Eiriksson [1], Gisli Masson[1], Unnur Thorsteinsdottir[1,4], Thorunn Rafnar [1], Patrick Sulem [1], Agnar Helgason[1,5], Daniel F. Gudbjartsson [1,3], Bjarni V. Halldorsson [1,2] ✉ & Kari Stefansson [1] ✉

Microsatellites are polymorphic tracts of short tandem repeats with one to six base-pair (bp) motifs and are some of the most polymorphic variants in the genome. Using 6084 Icelandic parent-offspring trios we estimate 63.7 (95% CI: 61.9–65.4) microsatellite de novo mutations (mDNMs) per offspring per generation, excluding one bp repeats motifs (homopolymers) the estimate is 48.2 mDNMs (95% CI: 46.7–49.6). Paternal mDNMs occur at longer repeats than maternal ones, which are in turn larger with a mean size of 3.4 bp vs 3.1 bp for paternal ones. mDNMs increase by 0.97 (95% CI: 0.90–1.04) and 0.31 (95% CI: 0.25–0.37) per year of father's and mother's age at conception, respectively. Here, we find two independent coding variants that associate with the number of mDNMs transmitted to offspring; The minor allele of a missense variant (allele frequency (AF) = 1.9%) in *MSH2*, a mismatch repair gene, increases transmitted mDNMs from both parents (effect: 13.1 paternal and 7.8 maternal mDNMs). A synonymous variant (AF = 20.3%) in *NEIL2*, a DNA damage repair gene, increases paternally transmitted mDNMs (effect: 4.4 mDNMs). Thus, the microsatellite mutation rate in humans is in part under genetic control.

Mutations enable life to evolve and adapt. Accurate estimates of the rate of mutations and the processes behind them are therefore imperative for understanding evolution, making inferences about population history[1–6] and understanding the genetics of disease and other phenotypes[7–11].

Around 3% of the human genome are short tandem repeats (STRs)[12], some of which are polymorphic, i.e. microsatellites, and mutate several orders of magnitude faster than non-repetitive sequences[13]. The forces that shape age related mDNM accumulation in the two sexes and the effects of parental genotypes on it is largely unknown.

Genetic factors responsible for genome integrity can be expected to play a role in DNM accumulation. Sequence variants that increase single nucleotide polymorphism (sDNM) and mDNM rates are known in somatic tissues in humans[14] and sequence variants that increase the sDNM rate are known in animal and yeast models[15,16]. Apart from a

[1]deCODE genetics / Amgen Inc., Reykjavik, Iceland. [2]School of Technology, Reykjavik University, Reykjavik, Iceland. [3]School of Engineering and Natural Sciences, University of Iceland, Reykjavik, Iceland. [4]Faculty of Medicine, School of Health Sciences, University of Iceland, Reykjavik, Iceland. [5]Department of Anthropology, University of Iceland, Reykjavik, Iceland. ✉e-mail: bjarni.halldorsson@decode.is; kstefans@decode.is

handful of clinical cases[17,18], no human germline mutators are known that affect de novo mutation rate of microsatellites or other types of variants. Thus, while variants causing increased somatic mutational burden and disease risk over the carrier's lifespan are known[19,20], no variants are known to increase the number of germline de novo mutations present in the offspring of the carriers.

Most mDNMs are believed to occur as a result of failure in the processes responsible for sequence fidelity during DNA replication. These processes verify the correct pairing of bp and replace incorrectly paired or damaged bases[21,22]. Loss of function mutations affecting genes responsible for sequence fidelity are known to cause somatic microsatellite instability, which in turn can result in increased risk of colorectal, gastric, endometrial and other types of cancer[23].

Recombination, DNA damage repair and nonhomologous end joining (NHEJ) have also been implicated as determinants of mDNMs[24,25] and since the two germlines of the sexes have different exposure to these processes it is logical to assume that the mDNM accumulation differs between them. Spermatogonia undergo mitosis continuously, increasing the risk of mDNMs due to replication errors. In contrast, oocytes have been through fewer mitotic divisions, but are subject to a higher recombination rate[26] and spend many years in dictyate arrest, where homologous chromosomes are under structural stress because they are connected via chiasmata and the oocyte cohesin complexes that bind them together deteriorate with age. The damage resulting from this is then repaired by NHEJ or homologous recombination[27,28] which can give rise to mDNMs. The rate of mDNMs in humans has been shown to vary by a number of microsatellite properties[29–31], parent of origin, and paternal[32] age, but researchers have to date been unable to detect a relationship with maternal age.

Previous studies of mDNMs have largely been confined to small sets of well-characterized microsatellites, focused on trio cohorts with affected offspring or on specific diseases[1,32,33]. Here, we identify and genotype microsatellites in two large sequencing cohorts with the aim of estimating polymorphism and mDNM rate and finding environmental and genetic determinants of mDNMs (Fig. 1).

## Results

### STR identification and genotyping

We used popSTR[34] to call genotypes at 5,401,401 autosomal short tandem repeats (STRs) in two large WGS data sets, 53,026 Icelanders and 150,119 participants in the UK Biobank (UKB), sequenced to an average coverage of 39.2× (min: 19.7, max: 608.3) and 32.5× (min: 23.6, max: 128.1), respectively.

The STRs were identified with Tandem repeats finder[35], (Methods). Each STR has a repeat motif along with start and end positions in the reference and we refer to the sequence between these positions as the reference repeat tract (RRT) and its length as RRT length. For motif lengths between one and three bp, we compared the behavior of motif equivalence classes where all members of a class are either a circular shift or a reverse complement of each other. For example, the members of the AAT motif equivalence class are: AAT, ATA, TAA, ATT, TAT and TTA (Supplementary Table 1).

We found 1,394,292 (25.8%) and 2,393,292 (44.3%) of the STRs to be polymorphic in the Icelandic and UK data sets, respectively and will refer to these polymorphic STRs as microsatellites. We describe microsatellite diversity through polymorphism rate (the fraction of STRs that are polymorphic) and expected heterozygosity[36], (Methods, Supplementary Figs. 1 and 2, Supplementary Tables 2–5). In both the Icelandic and UKB data sets, polymorphism rate and expected heterozygosity were correlated with the number of bp in the motif (motif length), RRT length, fraction of G/C bases in the STR's motif (motif GC content), and repeat purity (Methods, Supplementary Note 1). The direction and effect size of these attributes on both polymorphism rates and expected heterozygosity is consistent between the two data

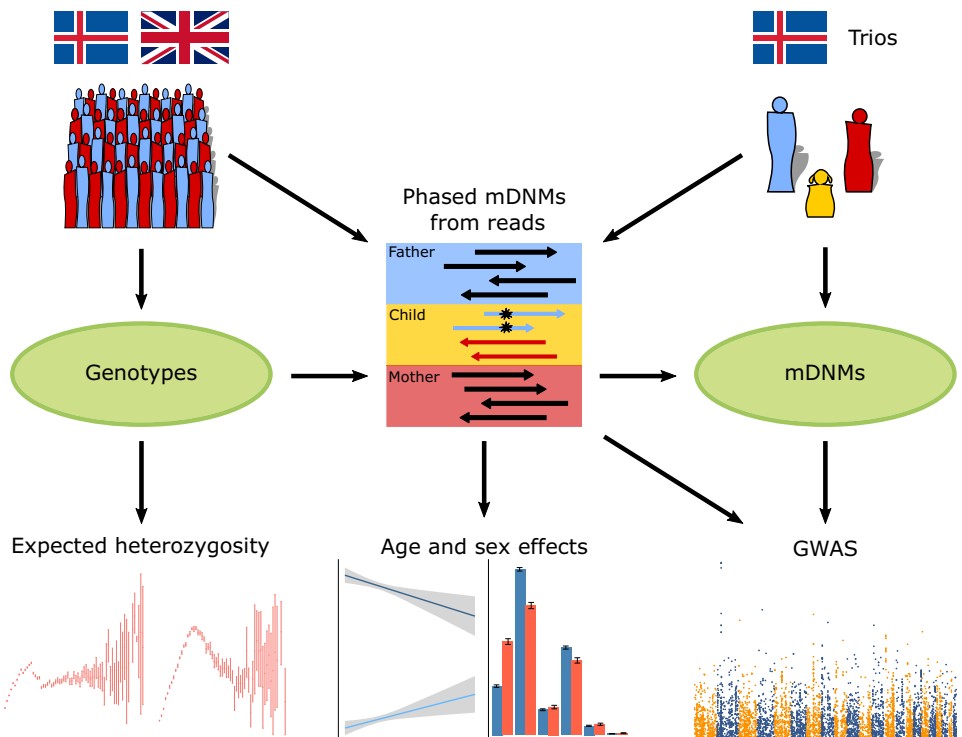

**Fig. 1 | Overview of the analysis.** We use WGS data from Iceland and UKB and genealogy data from Iceland. From the WGS data we generate microsatellite genotypes. Using trios in the genealogy and the genotypes we detect and phase mDNMs and count the number of mDNMs per trio. We associate the individual mDNM counts with genotypes of the parents in the trios. We compute population wide expected heterozygosity based on the genotypes and observe how it is affected by sequence context. From the phased mDNMs we also estimate age and sex effects on the mDNM rate and create parental phenotypes based on number of mDNMs found in offspring from the phased mDNMs.

sets but due to the greater size and genetic diversity of the UKB set, a greater number of microsatellites were found. Our microsatellite genotyping is limited by read length (151 bp for most samples), resulting in a decrease in accuracy when detecting and determining genotypes of alleles with long RRTs. While the RRT length limit for inclusion in our study is 140 bp, the average expected heterozygosity in our data sets decreases at RRT lengths exceeding 80 bp (Supplementary Fig. 1, Supplementary Fig. 2), so we conclude that this is the length where the effect of the RRT length on the genotyping accuracy becomes pronounced. We used different lower bounds for RRT depending on motif length (Supplementary Table 6). Further, microsatellites with one bp motifs, i.e. homopolymers, have higher sequencing error rates than others and therefore analyses of these have larger confidence intervals and less accuracy. In light of this, all joint motif length results are presented both with and without homopolymers.

## mDNM rate

We detected 76,987 mDNMs in 6084 Icelandic trios (mean: 12.7 mDNMs per trio) in 634,406 high-quality microsatellites (46% of the 1,394,292 microsatellites observed in the Icelandic data set), where the genotypes of the offspring were inconsistent with those of the parents (Methods). Each trio yielded on average 256,066 microsatellites (40.4%) where all three genotypes passed quality filters and we were able to test for mDNMs.

We estimated the false positive rate of our method between 2.0–5.6% using three approaches; PacBio CCS sequence data, mDNM sharing between monozygotic twins and haplotype sharing across three-generation families (Supplementary Note 2). PacBio CCS data were available for four of our trios, however we were unable to use them for verification of homopolymer mDNMs since the sequencing error rate was too high at these locations. Out of 27 mDNMs, we observed one false positive mDNM at a dinucleotide microsatellite while the other 26 were confirmed as true positives (Supplementary Table 7). Not surprisingly, homopolymers had the lowest validation rate for both mDNMs shared by monozygotic twins (88.7%, Supplementary Table 8) and transmission rates where 40% of homopolymer mDNMs were transmitted from probands to their offspring (Supplementary Table 9). Parental allelic drop out events are a potential source of false positive calls, where genotyping errors in parents cause transmitted variants to falsely look like mDNMs. Since the validation methods described above are not sensitive to allelic drop out, we counted mDNMs without parental read support for the de novo allele to estimate its contribution. We counted mDNMs both where less than 5% of reads and at most two reads displayed the allele as well as where no reads displayed the de novo allele (Table 1). Out of our 76,987 mDNMs, 90% have less than 5% and at most two parental reads supporting the de novo allele and 83.4% have no supporting reads in the parents. We note that a low frequency of de novo supporting reads in parents does not definitively indicate allelic dropout since sequencing errors are particularly common for STRs and the presence of these alleles can be due to slippage events in sequencing[37,38]. The DNMs could also be somatically present in the parent[39]. When a mutation occurs during the embryonic development of the parent some but not all cells will carry it. If germ cells are among the cell types carrying the mutation, it can be transmitted to the proband[40,41]. If cells carrying the mutation are also present in the parental sample submitted for sequencing, the mutation can be detected in the resulting WGS data, but it depends on the variant allelic frequency whether it is included in the parent's germline genotype[39–41].

We estimated the average mDNM rate across all motif lengths as $4.95 \cdot 10^{-5}$ mutations per microsatellite per generation (MMG), consistent with previous results (Supplementary Note 3). mDNM rates for microsatellites are typically estimated using polymorphic STRs, while the DNM rate for SNPs is typically estimated at all reliably characterized bp in the genome. Larger cohorts will necessarily have more polymorphic STRs, and the ones added relative to the smaller cohorts will be enriched for microsatellites with low mutation rates, leading to different estimates of the mDNM rate depending on the cohort size (Supplementary Note 3). We observed an order of magnitude difference in mDNM rates between motif lengths, ranging from $1.04 \cdot 10^{-5}$ MMG for hexanucleotide repeats to $1.07 \cdot 10^{-4}$ MMG for dinucleotide repeats (Methods, Table 1, Fig. 2) and between motif equivalence classes (Supplementary Note 4). Using motif length specific mutation rates and the average number of markers available for each trio to extrapolate to the full set of 1,394,292 microsatellites gives an expected number of 63.7 mDNMs (95% CI: 61.9–65.4) per offspring per generation (Methods) and excluding homopolymers, the expected number of mDNMs is 48.2 (95% CI: 46.7–49.6). This extrapolated number is comparable to the de novo mutation events at SNPs and indels detected through short read sequencing[39,42].

We find that mDNMs occur more frequently in late replicating regions(Methods, Supplementary Note 5), consistent with lower replication fidelity in those regions. mDNMs are rarer within functionally annotated regions and we also find that exon intersecting microsatellites contain more interruptions than intergenic ones (Supplementary Note 5). Since longer uninterrupted repeats are associated with higher mDNM rates[30], this suggests that variants interrupting the repeat structure are positively selected in exons as a way to reduce the impact of further disruption of exon sequences by mDNMs.

We further replicate known mutation rate behaviors such as an expansion bias at shorter RRTs and contraction biases at longer RRTs (Supplementary Note 6).

## Parent of origin effects

To determine the differences between the sexes in mDNM formation, we assigned a parent of origin to 46,171 (60.0%) of the mDNMs using a combination of three methods; read pair tracing, allele sharing and haplotype sharing in three-generation families (Methods). The concordance was above 93% between the three methods (Supplementary Tables 10–12). We found mDNMs from fathers (N = 35,501, 76.9%) to be 3.3 (95% CI: 3.2–3.4, chi squared test $P < 1 \cdot 10^{-320}$) times more common than from mothers (N = 10,670, 23.1%) (Supplementary Table 13).

Maternal and paternal mDNMs occurred with different probabilities at different RRT lengths, motif lengths, and motif equivalence classes. First, a larger fraction of maternal mDNMs occurred at tri-, penta-, and hexanucleotide microsatellites, whereas a larger fraction of paternal mDNMs occurred at di- and tetranucleotide microsatellites (Supplementary Table 14, Fig. 3, Supplementary Note 7). We note that whilevv maternal mDNMs also have a statistically significant larger fraction occurring at homopolymer microsatellites this might in part be due to their higher error rates. Second, the average number of bp affected by each mDNM was greater from mothers (3.4 bp, 95% CI: 3.3–3.4) than fathers (3.1 bp, 95% CI: 3.1–3.1) (Mann–Whitney $U$ test $P = 7.6 \cdot 10^{-8}$, Supplementary Table 15), consistent with previous results[32]. Stratifying by motif length revealed that maternal mDNMs affected a greater number of bp than paternal mDNMs on average at homopolymer, di- and tetranucleotide microsatellites (Table 2). Paternal mDNMs occurred at microsatellites with greater RRT lengths. Stratifying by motif length, we observed significant RRT length differences between the sexes at di-, tetra-, penta- and hexanucleotide microsatellites (Supplementary Table 16).

It seems likely that the sex differences in the accumulation of mDNMs transmitted to offspring are due to differences in the exposure to mutagens of the germlines leading to oocytes and spermatogonia. For di- and trinucleotide microsatellites, the relative frequency of maternal and paternal mDNMs differed between the motif classes (Supplementary Table 17). An enrichment of CCG motifs in maternal mDNMs may be the result of the vulnerability of GC rich sequences to

## Table 1 | mDNM rates and stats

| Motif (bp) | mDNM rate (95% CI) | #microsatellites | #DNMs | <2 parental reads and <5% parental reads | No parental reads |
|---|---|---|---|---|---|
| 1 | $2.15 \cdot 10^{-5}$ ($2.11 \cdot 10^{-5}$–$2.19 \cdot 10^{-5}$) | 399,087 (62.9%) | 17,194 (22.3%) | 14,875 (21.5%) | 12,982 |
| 2 | $1.07 \cdot 10^{-4}$ ($1.06 \cdot 10^{-4}$–$1.09 \cdot 10^{-4}$) | 93,653 (14.8%) | 33,025 (42.9%) | 28,771 (41.5%) | 26,259 |
| 3 | $4.92 \cdot 10^{-5}$ ($4.70 \cdot 10^{-4}$–$5.21 \cdot 10^{-4}$) | 29,869 (4.7%) | 5469 (7.1%) | 5156 (7.4%) | 4972 |
| 4 | $8.57 \cdot 10^{-5}$ ($8.28 \cdot 10^{-4}$–$8.86 \cdot 10^{-4}$) | 66,541 (10.5%) | 18,795 (24.4%) | 18,078 (26.0%) | 17,664 |
| 5 | $2.58 \cdot 10^{-5}$ ($2.37 \cdot 10^{-5}$–$2.86 \cdot 10^{-5}$) | 29,992 (4.7%) | 2132 (2.8%) | 2050 (3.0%) | 2012 |
| 6 | $1.04 \cdot 10^{-5}$ ($9.07 \cdot 10^{-6}$–$1.23 \cdot 10^{-5}$) | 15,264 (2.4%) | 372 (0.5%) | 337 (0.5%) | 322 |
| Total | $4.95 \cdot 10^{-5}$ ($4.88 \cdot 10^{-5}$–$5.02 \cdot 10^{-5}$) | 634,406 | 76,987 | 69,267 (90.0% of total) | 64,211 (83.4% of total) |

mDNM rate, number of microsatellites in high-quality set and number of mDNMs, all by motif lengths. Dinucleotide repeats have the highest mDNM rate and represent almost 43% of our mDNMs. The last two columns count number of mDNMs with less than 2 parental and less that 5% reads supporting the de novo allele and mDNMs with no parental reads supporting the de novo allele.

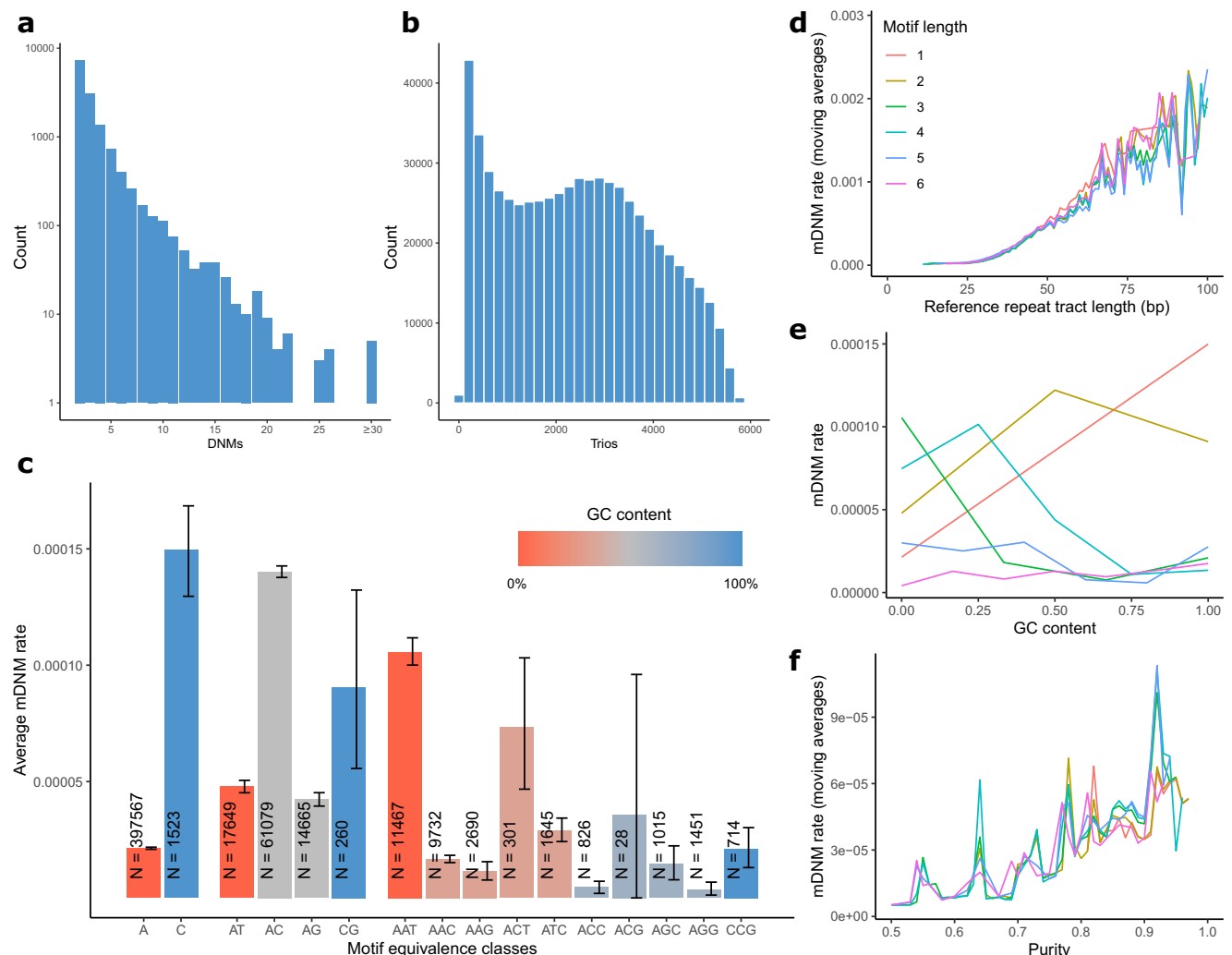

**Fig. 2 | mDNM rate. a** Histogram showing the number of mDNMs per micro-satellite. Only microsatellites with at least one mDNM are shown and the counts are on a log-scale. **b** Histogram showing the number of trios available to check for mDNMs per microsatellite with at least one mDNM. **c** mDNM rates of motif equivalence classes for motif lengths between one and three bp with error bars representing 95% confidence intervals, $N = 522{,}612$. **d** mDNM rate as function of reference repeat tract length stratified on motif length. The reference repeat tract length affects mDNM rates in a similar way for all motif lengths. **e** mDNM rate as function of GC content in motif. **f** mDNM rate as a function of repeat purity stratified on motif length. The mDNM rate increases with purity for all motif lengths.

alkylation[43] or oxidative damage[44] and the long time oocytes spend in dictyate arrest before meiosis. Building on this, we considered all non-trinucleotide mDNMs in microsatellites whose motif contained only G or C and found a 2.3 (95% CI: 1.7–3.7, $P = 0.03$) fold enrichment of maternal mDNMs indicating that purely GC repeats are more prone to

mutate in maternal than paternal germlines. Excluding homopolymers the enrichment is 5.3 (95% CI: 1.1–29.0, $P = 0.04$) fold.

Because of a known trend towards paternal expansion at an ATTCT repeat associated with spinocerebellar ataxia 10 (*SCA10*)[45,46], we examined motifs from this class specifically to

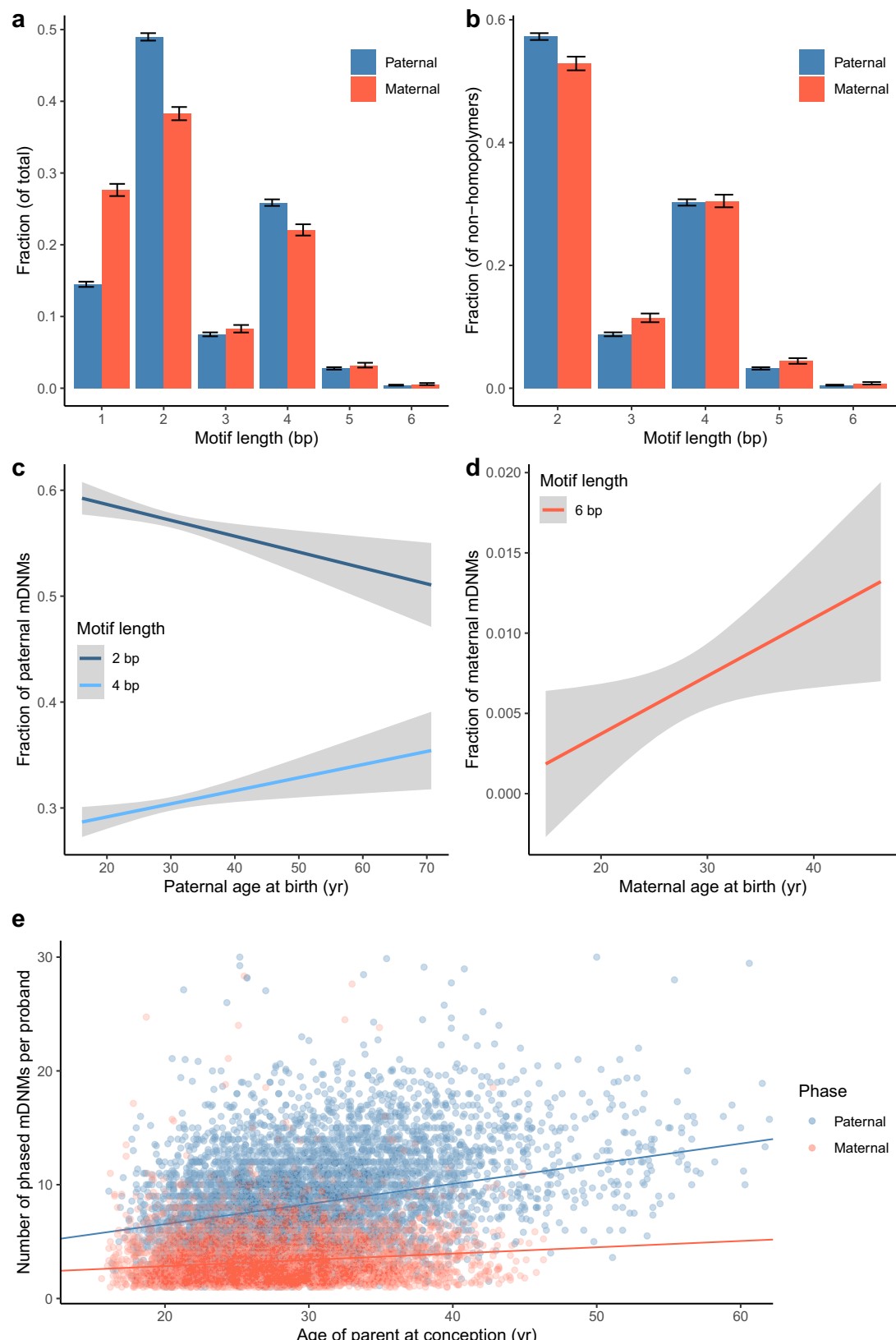

determine if the bias observed at *SCA10* could be observed genome-wide (Supplementary Table 18). The maternal contribution at pentanucleotide mDNMs was 25.9% overall, but for the ATTCT motif equivalence class, the ratio was only 18.9% (chi squared test $P = 0.041$). On average, we see an addition of 0.3 bp in paternal mDNMs at microsatellites with ATTCT motifs, compared

to a loss of 3.2 bp in maternal mDNMs (Mann–Whitney *U* test $P = 4.6 \cdot 10^{-4}$). This suggests that the kind of paternal expansion bias observed at the *SCA10* microsatellite is also seen for other ATTCT microsatellites in the genome. Further, for the myotonic dystrophy 1 associated CTG repeat in the *DMPK* gene, which is considered to have a maternal expansion bias[47], we do in fact see a

**Fig. 3 | Sex differences and age effects. a** Relative frequencies of paternal and maternal mDNMs at different motif lengths with error bars representing 95% confidence intervals.(nMaternal = 10,670, nPaternal = 35,501). **b** Relative frequencies of paternal and maternal mDNMs at different motif lengths without homopolymer mDNMs with error bars representing 95% confidence intervals.(nMaternal = 7776, nPaternal = 30,360). **c** Fraction of di- and tetranucleotide paternal mDNMs as a function of paternal age with error bars representing 95% confidence intervals. The tetranucleotide fraction increases while the dinucleotide fraction decreases. **d** The fraction of hexanucleotide maternal mDNMs as a function of maternal age with error bars representing 95% confidence. **e** Paternal and maternal age effect regression lines within our mDNM set. The age effect reported is interpolated to a genome-wide value using the fraction between average number of available microsatellites and total number of microsatellites. Fractions in (**c**) and (**d**) are computed after excluding homopolymers.

## Table 2 | Step sizes per parent

| Motif (bp) | Paternal | Motifs affected | Maternal | Motifs affected | Mann–Whitney *P* |
|---|---|---|---|---|---|
| 1 | 5141 (14.5%) | 1.67 | 2962 (27.6%) | 2.13 | **$1.9 \cdot 10^{-23}$** |
| 2 | 17,388 (49.0%) | 1.38 | 4119 (38.4%) | 1.71 | **$6.0 \cdot 10^{-13}$** |
| 3 | 2663 (7.5%) | 1.22 | 887 (8.2%) | 1.33 | $5.5 \cdot 10^{-2}$ |
| 4 | 9184 (25.9%) | 1.06 | 2365 (22.0%) | 1.12 | **$9.6 \cdot 10^{-5}$** |
| 5 | 977 (2.8%) | 1.08 | 343 (3.2%) | 1.21 | 0.38 |
| 6 | 148 (0.4%) | 0.95 | 62 (0.6%) | 0.77 | 0.26 |
| Total | 35,501 | 1.32 | 10,670 | 1.64 | **$1.1 \cdot 10^{-25}$** |

Motif length composition of paternal and maternal mDNMs, mean number of motifs added/removed for each motif length in each parent and one sided Mann–Whitney *U* test *p*-values for different step sizes between parents. Bold represents significant difference in step size between parents (*p* < 0.05).

## Table 3 | Parental age effect

| Data set | Paternal age effect (95% CI) | Maternal age effect (95% CI) | #paternal/#maternal |
|---|---|---|---|
| All markers | **0.178 (0.165–0.190)** | **0.058 (0.047–0.069)** | **35,501/10,670** |
| Motif length 1 | **0.013 (0.006–0.020)** | 0.007 (−0.001 to 0.014) | 5141/2948 |
| Motif length 2 | **0.078 (0.069–0.087)** | **0.029 (0.022–0.037)** | **17,388/4084** |
| Motif length 3 | **0.011 (0.008–0.014)** | 0.002 (−0.001 to 0.004) | 2663/883 |
| Motif length 4 | **0.056 (0.050–0.062)** | **0.013 (0.008–0.017)** | **9184/2354** |
| Motif length 5 | 0.004 (−0.003 to 0.011) | −0.002 (−0.007 to 0.003) | 977/341 |
| Motif length 6 | −0.008 (−0.021 to 0.005) | **0.016 (0.003–0.029)** | **148/60** |

Maternal and paternal age effect for all motif lengths and conditioning on motif length. mDNMs at mono-, di-, tri- and tetranucleotide repeats increase with paternal age while mDNMs at di- and tetra- and hexanucleotide repeats increase with maternal age. Bold represents significant association (*P* < 0.05 using a one sided $X^2$ test).

maternal bias (chi squared test *P* = 0.040) although without a significant difference in mutation size between the sexes (Mann–Whitney *U* test *P* = 0.05).

### Parental age influences number of mDNMs

The number of mDNMs transmitted to the offspring is affected by both paternal ($P = 5.4 \cdot 10^{-176}$) and maternal ($P = 7.2 \cdot 10^{-24}$) age at conception (Methods). The increase is 0.97 mDNMs per year in fathers (95% CI: 0.90–1.04) and 0.31 mDNMs per year in mothers (95% CI: 0.25–0.37) age, resulting in an expected number of 51.2 (35.7 paternal and 15.5 maternal) mDNMs and 77.0 (55.1 paternal and 21.8 maternal) mDNMs extrapolated genome-wide in offspring of 20- and 40-year-old parents, respectively. An increase of mDNMs with paternal age is consistent with previous studies[32], but, to our knowledge, this is the first demonstration that mDNMs are also affected by maternal age.

mDNMs at mono-, di-, tri- and tetranucleotide microsatellites increased significantly with paternal age and mDNMs at di-, tetra- and hexanucleotide repeats increase with maternal age (Table 3). An age effect is likely for all motif lengths in both sexes but we lack power to detect it in less frequent motif lengths.

When we consider the fraction of mDNMs of each motif length that each individual carries we see a clear difference in the mutagens acting on the two sexes (Supplementary Note 8). The fraction of dinucleotide paternal mDNMs decreased with paternal age, while the fraction of tetranucleotide mDNMs increased with paternal age (Fig. 3). The fraction of hexanucleotide maternal mDNMs increased with maternal age (Fig. 3). The heterogeneity between the sexes in motif length fractions that change with age indicates that, for the two sexes, the processes behind age related mDNM accumulation are likely to preferentially mutate microsatellites at different motif lengths.

### Parental genotypes associate with number of mDNMs in offspring

To determine whether there are variants in the genome that affect the number of mDNMs transmitted by parents to their offspring, we performed a genome-wide association study (GWAS) using the number of mDNMs originating from each parent as a phenotype, adjusting for age, sex, sequencing method and sample type (Methods). Here, we are not looking for variants that affect the accumulation of somatic microsatellite mutations during the carrier's lifespan, but rather for variants that affect the number of microsatellite mutations in the carrier's germ cells and can be transmitted to and detected as mDNMs in their offspring.

Three correlated SNPs rs4987188, rs112587140 and rs113983130 (Supplementary Table 19) were associated with an increase in mDNMs for all motif lengths. Based on sequence-annotation weighted Bonferroni-correction[48], we selected rs4987188[A], a missense variant (p.Gly322Asp, AF = 1.9%) in the gene *MSH2*, a mismatch repair gene[49,50], as the lead marker for the association (Fig. 4, Supplementary Table 20).

Each copy of rs4987188[A] is associated ($P = 3.6 \cdot 10^{-10}$) with a 0.37 s.d. (95% CI: 0.26–0.48) increase in the number of transmitted mDNMs, corresponding to 13.1 and 7.8 mDNMs genome-wide per

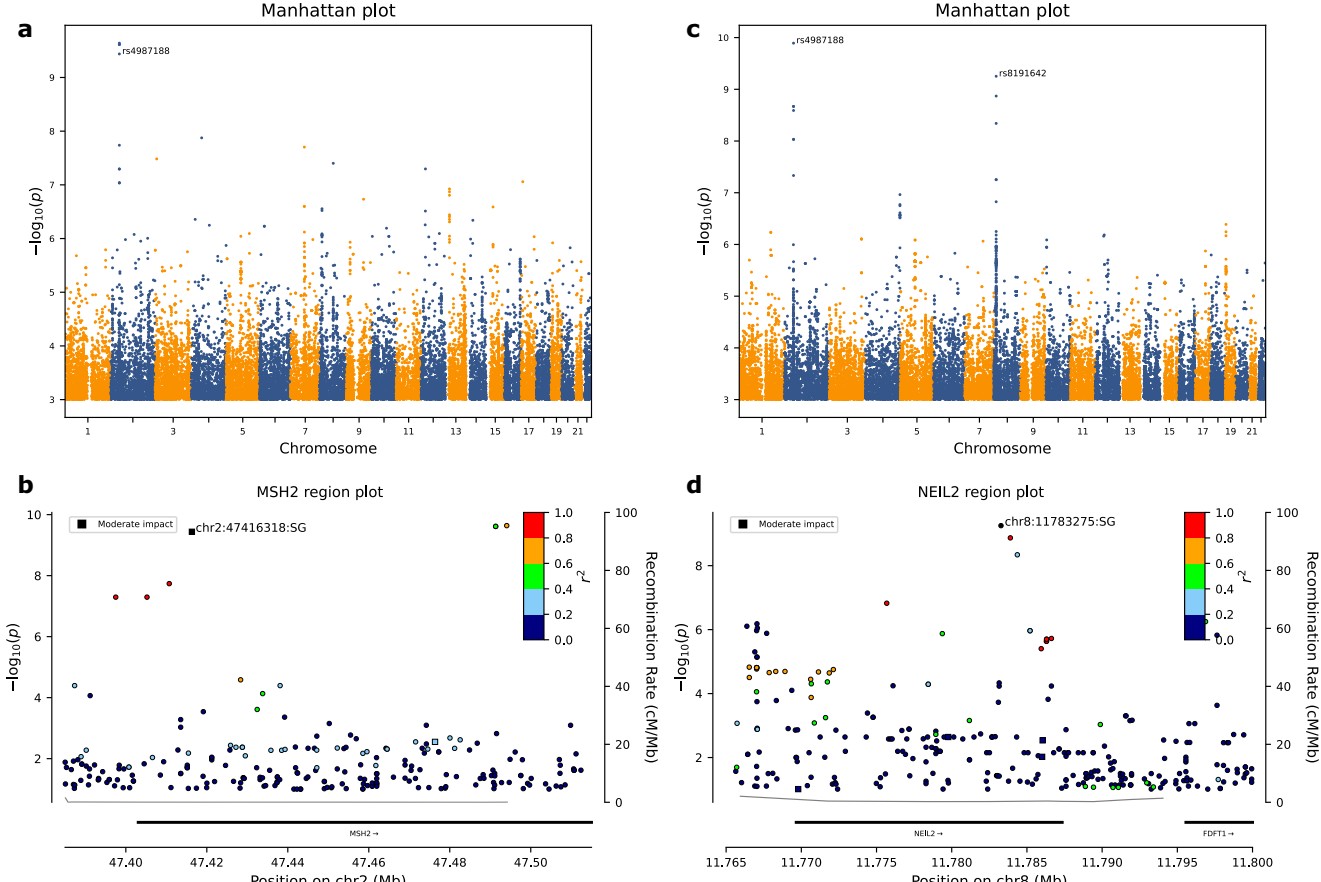

**Fig. 4 | Genome-wide association. a** Manhattan plot showing a missense variant in the MSH2 gene associating with an increased number of mDNMs transmitted to offspring. **b** Region plot for the missense variant in MSH2. Two correlated markers also reach genome-wide significance. **c** Manhattan plot showing a synonymous variant in the NEIL2 gene which associates with an increased number of mDNMs with motif lengths greater than one bp transmitted to offspring. The plot also shows the p-value for the MSH2 missense variant considering mDNMs with motif lengths greater than one bp. **d** Region plot for a synonymous variant in the NEIL2 gene. No other markers at the locus reach genome-wide significance. Chromosomes on Manhattan plots are marked with alternating colors and the threshold for genome-wide significance was set as $1 \cdot 10^{-9}$.

paternal and maternal copy, respectively. No significant difference in effect was found in the increase between the two parental sexes ($P = 0.14$). However, we see a nominal association ($P = 0.019$) between paternal age and mutation count in offspring of male carriers, suggesting an increase in the effect of rs4987188[A] with paternal age. Effect sizes and p-values for each motif length separately are given in Table 4.

The protein encoded by *MSH2* forms two heterodimeric mismatch repair complexes. One predominantly required for repairing

mismatched bp and the other mainly responsible for repairing insertion/deletion loops between one and twelve bp[51,52]. Multiple variants in *MSH2* have been reported to cause Lynch syndrome (also known as hereditary non-polyposis colorectal cancer, HNPCC), which results in increased risk of endometrial, colorectal and other cancers[53]. We tested rs4987188[A] for association with an increased risk of endometrial and colorectal cancer in both the Icelandic and combined meta-analysis data sets available at deCODE genetics, but found no such association. Functional studies of the yeast homolog of rs4987188 indicate that it results in a modest decrease in mismatch repair efficiency[54,55] (Supplementary Note 9).

Homopolymer mDNMs have a higher false positive rate than other mDNM categories. To assess the robustness of our results, we reran the GWAS on mDNM counts per parent without mDNMs occurring at homopolymers. The *MSH2* association was retained with a similar effect of 0.39 s.d. ($P = 1.3 \cdot 10^{-10}$). Furthermore, we found a second association between mDNM counts without homopolymers and two correlated ($r^2 = 0.988$) SNPs, rs8191642 ($P = 5.6 \cdot 10^{-10}$) a synonymous variant (Pro188), and rs8191649 ($1.4 \cdot 10^{-9}$), an intronic variant, both in *NEIL2*, a glycolase involved in both transcription and replication associated base-excision repair of DNA damaged by oxidation or by mutagenic agents[56]. We selected[48] rs8191642[G] (AF = 20.3%), as the lead marker for the association (Fig. 4).

Testing rs8191642[G] for mothers and fathers separately revealed heterogeneity in effect between the sexes ($P = 1.2 \cdot 10^{-3}$). Each copy of rs8191642[G] is associated with ($P = 7.2 \cdot 10^{-12}$) a 0.21 s.d. (95% CI:

## Table 4 | GWAS effects and *p* values

| Motif length | Effect(s.d.)/*P* (rs4987188 in *MSH2*) | Paternal effect(s.d.)/*P* (rs8191642 in *NEIL2*) |
|---|---|---|
| 1 | 0.14/2.0 · 10⁻² | 0.20/1.8 · 10⁻² |
| 2 | **0.31/2.1 · 10⁻⁷** | **0.44/1.2 · 10⁻⁷** |
| 3 | 0.07/0.25 | 0.11/0.18 |
| 4 | **0.26/1.4 · 10⁻⁵** | **0.23/5.5 · 10⁻³** |
| 5 | 0.03/0.59 | 0.03/0.74 |
| 6 | −0.01/0.87 | 3 · 10⁻³/0.97 |
| 2-6 combined | **0.39/1.3 · 10⁻¹⁰** | **0.21/7.2 · 10⁻¹²** |
| All combined | **0.37/3.6 · 10⁻¹⁰** | **0.17/2.3 · 10⁻⁶** |

Effect sizes and p-values at each motif length separately, for motif lengths between two and six bp and all motif lengths combined for rs4987188 and rs8191642. Bold represents genome-wide significant association effects after correcting for multiple testing (Threshold of $1 \cdot 10^{-9}$ for initial detection and $6.3 \cdot 10^{-3}$ (0.05/8) for per motif length tests).

0.15–0.27) increase in the number of paternally transmitted mDNMs with motif lengths greater than one bp, corresponding to 4.4 mDNMs genome-wide per copy, but associated with a modest increase in the number of maternally transmitted mDNMs ($P = 0.015$, effect = 0.072 s.d.). Effect sizes and *p*-values of paternal carriers for each motif length separately, for all motif lengths combined and excluding homopolymers are given in Table 4.

The glycolase encoded by *NEIL2* participates in both transcription and replication coupled repair and as rs8191642 is synonymous, it may act through expression although we are not aware of any functional studies on the variant or its homologs. We looked at the association between rs8191642 and RNA expression of *NEIL2* and found that each copy of rs8191642[G] is associated with ($P = 3.0 \cdot 10^{-412}$) a 0.57 s.d. decrease in the expression of *NEIL2*. We note that after correcting for two expression quantitative trait loci (eQTL) for RNA expression of *NEIL2* (rs17153755 and rs1293299), the association is no longer significant ($P = 0.49$). After correcting for these two eQTLs, the association of rs8191642 with an increased number of paternally transmitted mDNMs however remains significant ($P = 6.6 \cdot 10^{-7}$, effect = 0.22 s.d.) despite the increase in the *p*-value.

To assess the effect of these microsatellite mutator alleles on other types of genetic variation, we compared sDNMs transmitted from carrier and non-carrier mothers and fathers. The results of these comparisons suggest that the effects of rs4987188[A] and rs8191642[G] are confined to microsatellites. If they do affect the fidelity of the mismatch or base-excision repair pathways (Supplementary Note 10), then their effect is too small to be detected in our cohort. We also computed the correlation between the number of transmitted sDNMs and mDNMs for parents where we had counts for both (Supplementary Table 21) and found the correlation to be low (Paternal $R^2 = 0.03$ ($P = 0.32$), Maternal $R^2 = -0.04$ ($P = 0.2$)), consistent with results reported by others[57].

Finally, we used QuickGO[58] to identify a set of genes participating in mismatch repair, base-pair excision repair and nucleotide excision repair and tested whether coding variants within these genes were enriched for associations with our phenotypes. We tested for association enrichment in the six different phenotypes described above, measuring the number of transmitted mDNMs in all parents, fathers only and mothers only, both including and excluding homopolymers. After correcting for the number of genes tested, we observe a nominal enrichment of associations for all six phenotypes in *NEIL2* and for one or more of the phenotypes in several other genes (Supplementary Data 1).

## Discussion

We generated two large microsatellite genotype sets. The microsatellite genotypes for the UKB samples are publicly available and have been tested for association with various phenotypes[59].

In the Icelandic set, we identified 76,987 mDNMs and found an association between the number of mDNMs transmitted to offspring and the age of both mothers and fathers at conception and parental genotypes. We observed a previously unreported increase in the number of maternal mDNMs transmitted to offspring with maternal age, consistent with the increase of both SNP/indel DNMs[39] and recombination with maternal age[26,60]. mDNMs are often associated with misaligned DNA strands, formed transiently during DNA synthesis[29,30,61–64], however the maternal age effect gives novel insight into the formation of mDNMs, as oocytes remain in dictyate arrest until shortly before ovulation occurs[39], compared to the actively dividing spermatogonia in aging fathers. The maternal age effect indicates that, mDNMs can also occur outside of DNA synthesis during S-phase replication, presumably since DNA polymerases operate during most types of DNA repair on long tracts and in homologous recombination pathways[29].

The observation of different frequencies of mDNM motif classes transmitted by older mothers and fathers allows us to shed light on the mutational processes acting in the germlines. Since spermatogonia undergo a greater number of mitotic cell divisions than oocytes, we propose that the paternal enrichment of AC motif class mDNMs is due to out-of-register realignments during replication[63], whereas the maternal enrichment of mDNMs at pure GC repeats is likely to be a result of damage accumulated during the dictyate arrest of oocytes[26,28]. Sex differences between the replication and repair processes, other than replication frequency and time spent in dictyate arrest remain possible as causes for the sex differences we observe in mutational patterns, although we have found no evidence to support this hypothesis.

Sequence variants detected in humans that decrease sequence stability have for the most part been very deleterious and under strong negative selection[65]. Interestingly, the microsatellite mutators at *MSH2* and *NEIL2* identified here are both present at high frequencies and have large effects, thereby indicating that a genome-wide increase in the mDNM rate is not sufficiently deleterious to result in strong selection against these variants. The similar effect of the missense variant in *MSH2* across the sexes, indicates that gametes from both sexes are subject to the same sequence fidelity maintenance process. *NEIL2* has been reported to function in both transcription and replication coupled repair[56] and thus, it is likely that the association between the synonymous variant in *NEIL2* and the number of paternally transmitted mDNMs is due to the more frequent replication of spermatogonia.

While we acknowledge that our set contains 6000 trios coming from a single population we feel that since humans are a relatively homogenous species[66], Icelanders are likely to be representative. This has been demonstrated on multiple occasions with GWAS results from Iceland[67–70] and estimation of SNP/indel de novo mutation rates using Icelandic trios[39].

We have identified germline variants, segregating at high frequencies, that directly affect the mDNM rate in humans. We have also demonstrated that the number of maternally transmitted mDNMs increases with maternal age. Last, we have generated a publicly available microsatellite genotype set for 150,119 samples, a valuable resource for the scientific community in its efforts to better understand and define the many ways that microsatellites affect human phenotypes.

## Methods

### Ethics statement

Blood or buccal samples were taken from individuals participating in various studies, after receiving informed consent from them or their guardians. All sample identifiers were encrypted and all processing of personal data was in agreement with the conditions set by the Icelandic Data Protection Authority (PV_2017060950ÞS). Approval for these studies was provided by the National Bioethics Committee. The North West Research Ethics committee reviewed and approved the UKB's scientific protocol and operational procedures (REC Reference number: 06/MRE08/65). Data for this study were obtained and research conducted under the UKB applications license number 68574.

### Identification of tandem repeats

To generate a set of STR locations we ran version 4.09 of Tandem repeats finder[35] on the GRCh38[71] human reference genome with the following parameters:./trf409.linux64 genome.fa 2 7 7 80 10 22 100 −d −h −ngs > trf_out_100

### Purity computation

We define the ratio between the observed repeat units in a STR and the number of expected repeat units in a perfectly pure STR as the repeat

purity.

$$\text{purity} = \frac{\text{observed repeats}}{E[\text{repeats}|100\% \text{ pure}]} \quad (1)$$

### Expected heterozygosity computation

We compute expected heterozygosity using the following formula[36]

$$\text{gene diversity} = \left[\frac{n}{(n-1)}\right]\left(1 - \sum_{i=1}^{k} p_i^2\right) \quad (2)$$

where $n$ is the total number of sequences, $k$ is the number of alleles at the marker and $p_i$ is the frequency of each allele.

### Genotype and marker filtering

For a trio comparison we require all members of the trio to have a phred scaled genotype quality value higher than 60. We removed microsatellites which imputed to a 0% frequency and ones that failed our best practices filters. These filters consider coverage, genotype quality, number of samples genotyped and fraction of reads not supporting the reported genotype and are described in[59].

We further removed microsatellites outside the Tier 1 regions defined by GIAB[72] and microsatellites lying within CNV regions annotated by CNVnator[73].

After using long range phased SNP genotypes to phase and impute our microsatellite genotypes into the Icelandic population[74], microsatellites with alleles showing a strong deviation from the Hardy Weinberg equilibrium ($P < 1 \cdot 10^{-6}$) or a frequency weighted imputation information lower than 0.9 were removed[75]. We also required the frequency weighted phasing information to be greater than 0.6 and the $r^2$ for leave one out cross validation of phased genotypes to be greater than 0.5.

We looked for offspring with homozygous mDNMs less than 1Mbp apart, implying a haploid state, and checked them for large CNVs covering the region, causing autozygosity and spurious mDNM calls. Last, we crossed all mDNMs with deletions and duplications imputed into the offspring[76] and removed the ones intersecting.

### De novo detection

For marker-trio combinations where all conditions described above are met, we define a mDNM as a trio genotype combination satisfying neither of the following logical statements:

(1)  $\text{proband}_{A1} \in M \text{ and } \text{proband}_{A2} \in F$
(2)  $\text{proband}_{A1} \in F \text{ and } \text{proband}_{A2} \in M$

where $\text{proband}_{A1}$ and $\text{proband}_{A2}$ refer to the two alleles carried by the offspring and $M$ and $F$ define the allele pairs carried by the mother and father, respectively.

### Microsatellite attribute regression on mDNM rate

We performed a Poisson regression in R using the built in glm function, setting the family parameter as 'poisson' and using the number of available markers per trio as an offset. We ran the regression on both the full data set and stratified by motif length to examine if the effects of other attributes on the mDNM rate remained consistent across motif lengths. The script used for the analysis, attributeRegression.R, as well as the input, mutRateDataAll, are available in our github repository.

The direction and statistical significance for both RRT length and repeat purity remain consistent for all motif lengths but the effect of GC content in repeat motif is positive for homopolymer repeats (Supplementary Table 22).

### Obtaining confidence intervals for mDNM rate estimates

We used the boot package for R[77,78] to obtain confidence intervals for both our genome-wide mDNM rate estimate and the motif length specific estimates. 100 replicates, generated in the boot package by subsetting the data, were used in all cases and 95% confidence intervals extracted using the resulting quantiles. Our input contained one entry for each microsatellite, containing details of its attributes, the number of trios available for mDNM detection and the number of mDNMs detected. The script used for the analysis, mutRateCIs.R, as well as the input, mutRateDataAll, are available in our github repository.

### Extrapolation from mDNM rate to expected number of mDNMs

The per motif length mDNM rate was determined from our set of high-quality microsatellite calls. We estimated the expected number of per motif length mDNMs among all microsatellites by multiplying the mDNM rate and the number of microsatellites per motif length. The total expected mDNMs is the sum over all motif lengths

$$E[\text{mDNMs}] = \sum_{i=1}^{6} r_i \cdot n_i \quad (3)$$

where $r_i$ is the mDNM rate at motif length $i$ and $n_i$ is the genome-wide number of microsatellites with motif length $i$.

### mDNM overrepresentation analysis

We counted mDNMs within and outside of regions with various annotations and computed overrepresentation (OR) for each annotation type using the following equation

$$\text{OR} = \frac{\text{mDNMs}_a / \text{bp}_a}{\text{mDNMs}_{out} / \text{bp}_{out}} \quad (4)$$

Where $\text{mDNMs}_a$ enumerates the number of mDNMs within the annotated regions, $\text{bp}_a$ represents the total size of the annotated region in bp, $\text{mDNMs}_{out}$ enumerates the number of mDNMs outside of the annotated region and $\text{bp}_{out}$ represents the number of bp outside the annotated region.

### Microsatellite mDNM phasing

We determined the parent of origin of mDNMs, using three distinct methods; read pair tracing, allele sharing and haplotype sharing in three-generation families.

First, we used long range phased[74] SNP and indel genotypes available at deCODE genetics[74] to phase reads reporting mDNMs when possible. Read phases enabled us to assign a parent of origin to mDNMs according to the phase of the reads reporting the event since a read phased to one parent and reporting a mDNM indicates that the mDNM was transmitted from that parent. Not all mDNMs had supporting reads containing phased markers and for these, assigning a parent of origin was not possible (Supplementary Fig. 3).

Second, for trios where the de novo allele was seen in neither parent and the other offspring allele was only seen in one parent, we phased the mDNM to the other parent (Supplementary Fig. 4).

Last, we used haplotype sharing in three-generation families such that if the mDNM was transmitted from offspring to their children, we phased the de novo to the parent sharing a haplotype with its grandchild, and to the parent not sharing a haplotype if de novo was not transmitted (Supplementary Fig. 5).

### Parental age effect regression

To determine the effect of parental age at conception on the number of mDNMs transmitted to the offspring, we applied a Poisson regression model described in detail in[39]. In summary, the model starts by integrating out the number of available markers in each trio. The following function is then maximized to determine the paternal age

effect:

$$L(\alpha,\beta;y_P,y_M,y_U) = \sum_{y_P^* = y_P}^{y_P + y_U} f(y_M^* = y_T - y_P^* \alpha_M + A_M \beta_M) \times f(y_P^*; \alpha_P + A_P \beta_P)$$

(5)

Here P and M represent paternal and maternal, respectively. The function for the maternal age effect is identical with the roles of P and M reversed. $L$ denotes the Poisson likelihood function to be maximized, the age of parents at conception is denoted as $A_P$ and $A_M$, $\alpha_P$ and $\alpha_M$ represent the intercepts of the age effect, $\beta_P$ and $\beta_M$ are the slopes of the age effect, $y_P$, $y_M$ and $y_U$ denote the number of paternally phased, maternally phased and unphased mDNMs, respectively and $y_T$ represents the total number of mDNMs detected. Since some mDNMs are unphased, we refer to the latent true number of paternal and maternal mDNMs as $y_P^*$ and $y_M^*$, respectively.

This function was maximized using the R script em.R, available on our github page, using the nonlinear optimization function nlm. We built our choice of starting points for the optimization on the results presented in[39]. As in[39], we assessed the significance of an age effect by fitting a nested model, setting either $\beta_F$ or $\beta_M$ to 0, and evaluating the log likelihood difference between the full and the nested model with a $X^2$ approximation.

To determine whether the coefficients were robust to the phasing method, we repeated the analysis[39] and split the phased mDNMs by their phasing method (3 generation, read-tracing and allele-base) and performed the regression on each subset. Both maternal and paternal age remained significant in all subsets (Supplementary Table 23).

We compute the total predicted number of de novo mutations in an offspring with an $X$ year old father and a $Y$ year old mother using the coefficients from our regression model:

$$(I_F + \beta_F \cdot X + I_M + \beta_M \cdot Y) / \left( \frac{\mu_{\text{markers}}}{1,394,292} \right)$$

(6)

Where $I_F$ is the paternal intercept, $\beta_F$ is the paternal age effect, $I_M$ is the maternal intercept, $\beta_M$ is the maternal age effect, $\mu_{\text{markers}}$ is the mean value of available markers across our trios and 1,394,292 is the total number of microsatellites we detect.

The script used for the analysis, em.R, as well as the input, totalPerPn, are available in our github repository.

### Construction of mDNM phenotypes

We used maternal and paternal counts at each offspring to construct phenotypes for the parents quantifying the number of mDNMs passed on to the offspring. In addition to correcting for the parental sex (mother/father), we corrected for parental age at birth of offspring, sequencing method, sample type and number of microsatellites available for de novo detection in the trio. For parents with more than one offspring in our trio set, we used the average of all their offspring. After regressing out the coefficients, we used rank inverse normalization to normalize the phenotype.

### Association enrichment test in DNA repair genes

We used the refSeq annotations from ensemble 109 to obtain gene coordinates. We used only SNPs and small indels, all with imputation info values >0.9 and no genotyping quality issues and randomly removed one member of variant pairs with $R^2$ values >0.9. We counted the total number of exonic variants in the genes and then how many of those had association $p$-values <0.05. Last, we used the binom.test function in R to obtain association enrichment $p$-values.

### Reporting summary

Further information on research design is available in the Nature Portfolio Reporting Summary linked to this article.

### Data availability

Access to these data is controlled; the sequence data cannot be made publicly available because Icelandic law and the regulations of the Icelandic Data Protection Authority prohibit the release of individual-level and personally identifying data. Data access can be granted only at the facilities of deCODE genetics in Iceland, subject to Icelandic law regarding data usage. Anyone wishing to gain access to the data should contact K.S. (kstefans@decode.is) with a timeframe of one month for a response. Results from the association analysis to our phenotypes will be uploaded to the deCODE genetics website www.decode.com/summarydata. A list of the mDNMs generated and used in the study is available here: https://doi.org/10.5281/zenodo.8005262[79]. We have also uploaded a list of the markers considered and mDNM counts per marker to the same github repository[79]. WGS, genotype data, phased and imputed data for the UKB set can be accessed via the UKB research analysis platform (RAP): https://ukbiobank.dnanexus.com/landing. The Research Analysis Platform is open to researchers who are listed as collaborators on UKB-approved access applications. The UKB microsatellite genotypes were created as a part of this study and also presented in[59]. This research has been conducted using the UK Biobank Resource under Application number 68574.

### Code availability

We used popSTR (https://github.com/DecodeGenetics/popSTR v2.0) to generate microsatellite genotypes. Scripts used for analysis, as well as their input, are available here: https://doi.org/10.5281/zenodo.8005262.

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

## Acknowledgements

We thank our colleagues from deCODE genetics/Amgen Inc. We also thank all research participants who provided a biological sample to deCODE genetics and the UK Biobank.

## Author contributions

Paper was written by S.K, supervised by B.V.H, with input from H.J, O.A.S, G.A.A, U.T, T.R, A.H, P.S, D.F.G and K.S. S.K, H.P.E, A.G, P.I.O, G.M and B.V.H developed analysis tools, with input from H.J, O.E and D.F.G. G.P performed mDNM enrichment analysis. A.G and D.B generated SV genotypes. G.S performed gene burden test. G.H assembled and haplotype tagged PacBio HiFi data. Association analysis was performed by S.K, T.R and B.V.H. Age effect regression model was implemented by H.J. Expression analysis and association was performed by G.H.H. SNP mutation signatures were created by S.O. SNP genotyping was performed by H.P.E and P.I.O. Microsatellite genotyping was performed by S.K. Figures were drawn by M.T.H and S.K. Study was supervised by B.V.H and K.S. All authors agreed to the final version of the manuscript.

## Competing interests

All authors are employees of deCODE genetics/Amgen.
