## [Peer Review File · Nature Communications]

Sequence variants affecting the genome-wide rate of germline microsatellite mutationsREVIEWER COMMENTS

Reviewer #1 (Remarks to the Author):

This manuscript presents an interesting preliminary analysis of microsatellite mutations in a large set of trios. Echoing findings about single nucleotide polymorphisms, the authors find positive correlations of microsatellite mutation rates with paternal and maternal age, and they also find evidence that genetic variation at at least two loci regulates germline microsatellite stability. The existence of these genetic variants, in particular, will help others interpret variation in microsatellite variation within the human population, since such variation has been studied for decades but always under the assumption that mutation rates are constant between genotypes and populations.

Although the paper's results seem interesting and plausible, the paper is hindered by a lack of methodological detail. In some cases, key details are supplied in the supplement, but are omitted from the main text in a misleading or confusing way. In other cases, the underlying analysis may not be rigorous enough for the results to pass muster.

One weakness of the paper with potentially far reaching downstream consequences is a failure to estimate an accurate false positive rate for homopolymer mutations. The authors note that even HiFi data has a high error rate at homopolymer loci, and short read data certainly has a higher error rate at these loci compared to repeats of motif length greater than 1. The transmission rate of 0.4 at homopolymer mutation sites suggests that at least 20% of homopolymers calls are likely to be false positives. The number of false positives is actually higher than this in all likelihood, since third generation transmission is unable to detect false positives that are caused by allelic dropout, i.e. genotyping errors in the parents that cause variants transmitted to the offspring to falsely look like de novo mutations. Allelic dropout also invalidates the premise of the monozygotic twins analysis, since a genotyping error in one of the parents of a set of twins may cause the same false positive de novo mutation to be called in both twins.

To tighten up their validation scheme, the authors need to discuss the possibility of allelic dropout and attempt to minimize it perhaps by requiring that a variant called as a de novo mutation in a child has no supporting reads in either parent. They also need to discuss the results of the validation in the main text (including the percentage of each type of variants that validate by each method) rather than confining the results of these tests to the supplement. Finally, they need to mention much earlier in the manuscript that homopolymer mutations have poor validation rates, disclose how many of their calls are homopolymer mutations, and put error bars around all mutation rate estimates that reflect the high error rate of homopolymer mutations. Could false positive homopolymer mutations be responsible for the apparently enrichment of maternal mutations for homopolymer variants?

Another analysis that is lacking in methodological detail is the fine mapping component of the GWAS. The authors never explain how they narrow down their first peak to a single SNP in MSH2, or how they narrow down their second peak to a synonymous SNP in NEIL2 (which being synonymous seems unlikely to have a large functional effect). My impression is that the typical GWAS peak is associated with quite a wide confidence interval containing several genes. I assume that the authors picked MSH2 and NEIL2

from within larger confidence intervals because of these genes' prior associations with microsatellite instability, which is reasonable, but the paper needs to be more transparent about how the selection was done, how many other genes are within the peak confidence interval, and whether any of them have functions related to DNA repair or other processes that could impact microsatellite mutation rates.

Finally, given that the central point of this paper is to measure a microsatellite mutation rate, I was struck by the fact that the abstract does not actually mention a mutation rate, but only a raw count of de novo mutations per genome per generation. At several points, the paper mentions "high quality" microsatellites and alludes to the fact that the authors only call mutations at a subset of high quality loci, but this is not discussed at length enough to make it obvious how the raw number of mutations per genome per generation is translated into a mutation rate per locus. A supplementary note also mentions that the sample size of the cohort affects the mutation rate measurement, which seems counterintuitive and is not something that is true of SNV mutation rate measurements. I think it would be more appropriate for the main text to discuss estimation of a mutation rate per locus per generation in a manner that is comparable to other studies, given that this is a central challenge addressed by papers that estimate single nucleotide de novo mutation rates.

Reviewer #2 (Remarks to the Author):

The manuscript by Kristmundsdottir et al. used a novel Icelandic parent-offspring trio WGS dataset to estimate the incidence of de novo microsatellite mutations (mDMNs) in humans. Their analysis was inclusive of 1-6 bases/unit microsatellite classes (e.g., mononucleotide through hexanucleotide repeats). The first part of the study confirms the impact of inherent DNA variables (length, sequence, purity) on microsatellite mutation that has been well established by numerous previous studies of humans (e.g., 1000 genomes), primates, and other organisms. From this foundation, the authors examined the association of mDMNs with parental age and sex to deduce additional factors affecting microsatellite mutation. Finally, the authors performed a GWAS analysis to discover SNPs associated with mDMN incidence.

In general, the methods used are straight-forward and add important knowledge regarding germline rates of microsatellite mutation in humans. Parental biases in microsatellite mutation have been documented for a few microsatellites that cause disease, and this study now provides a genome-wide analysis of parental biases (sex and age) which is an important advancement. However, while the GWAS section is potentially the most innovative and biologically significant section of the study, insufficient evidence is provided to support the stated major conclusion that this study shows the "microsatellite mutation rate in humans is in part under genetic control". The authors overinterpret and overstate the significance of the GWAS results, and additional data are required to support the conclusion that the identified variants in MSH2 (rs4987188) and NEIL2 (rs8191642) are "microsatellite mutators". The authors state (lines 211-215; Supplementary note 9) that their results of an association between the variants and mutation are confined to microsatellites, as they found no association with base

substitution mutational signatures. If true, this would be an unprecedented result, because no genetic mutation that solely impacts microsatellite mutation without impacting single nucleotide substitutions or indels has been demonstrated in over 2 decades of microsatellite mutation research. The alternative explanation of their results is that the SNP associations have no mechanistic impact on mDNMs. Because of this, it is imperative that the same results be shown by another approach (for instance, an experimental model system).

Additional concerns and suggestions for improvement are provided in detail below.

1. Association of mDMN incidence with DNA repair gene SNPs. This is the weakest section of the study. The MSH2 missense variant has been studied in a yeast model (references 46 and 47). In these publications, it was shown that the variant is functional (able to complement an MSH-deficient cell), and the authors of ref. 47 conclude that any impact on mutation rates would be due to levels of protein expression. In the current study, Kristmundsdottir et al. provide no data showing that expression of the MSH2 variant protein is altered, compared to wild-type. Furthermore, they found no association of the variant allele with cancer risk (as would be expected for a defective MSH2 allele), suggesting the variant is functioning normally in maintaining genome stability. The association with the NEIL2 variant is not compelling. This is a synonymous variant and no literature exists to demonstrate this SNP alters NEIL2 function. Furthermore, no evidence is provided by the authors or cited from the literature to support a role for NEIL2 and base excision repair in microsatellite mutagenesis.

2. The manuscript's methods should be improved to provide more transparency and allow others to reproduce the results. For the RRT, did the authors use a lower limit for the number of repeat units to define a microsatellite (e.g., were 2 unit motifs as in GG included)? Was this lower limit the same for all units (mono, di, tri, etc)? This is important, as published microsatellite analyses use different definitions. At the other end, was there an upper limit to the RRTs? The authors imply (lines 85-88) that genotypes above 80 bp were unreliable, so was 80bp the upper limit for all STRs? Lastly, lines 463-464 state that WGS regions were removed/excluded. The authors should provide the number of bases or % of the whole genome sequence that were actually included in the analyses.

3. The authors discuss only one possible mechanism underlying the sex bias observed, that oocytes sustain more DNA damage than sperm. They state "It seems likely", but provide no scholarly references to support their hypothesis that oocyte damage is greater than sperm damage. The rationale for their hypothesis needs to be more developed, and the paper would be strengthened by citing publications showing that oxidative or alkylation damage increases microsatellite mutation. The alternative hypothesis, that DNA replication or DNA repair programs differ in oocytes and sperm, are equally plausible and should be discussed. Also, parental biases in microsatellite mutation have been well documented for microsatellite expansion diseases. The authors should also cite biases known from the literature (in addition to SCA10). For instance, DM-1 (myotonic dystrophy) shows a paternal bias, while FRAX mutations show a maternal bias.

4. The results showing the size of maternal versus paternal mDNMs are not clearly presented and it is difficult to independently assess the authors' conclusions. The authors state in the Abstract (line 22) that "maternal mDNMs affect more bp", a statement that is ambiguous. What does "more bp" mean, and how was this calculated? Does the "more" mean a comparison to parental mDNMs? Table 3 presents the size of mDNMs from each parental sex, by microsatellite class. However, the conclusion stated in the table legend and in the Results (lines 124-126) that "maternal mDNMs are larger on average than parental ones" is not apparent in the data presented.

5. The biological significance of the age effects measured should be discussed. The authors state that mDNMs increase by 0.31/year for maternal age at conception. Given the human window for conception spans 30 years (~15 yoa to 45yoa) on average, this corresponds to an increase, at most of 10 mDNMs due to maternal age, or ~15 % of the overall rate stated (64 mDNMs/offspring/generation).

6. Introduction (lines 51-59) states that "apart from a handful of cases, no germline mutators are known". The authors have failed to discuss the well known constitutional mismatch repair deficiency, caused by homozygous inheritance of mutations in mismatch repair genes. These references should be included in the manuscript.

7. The authors should discuss the limitations of this study. Findings from a population of ~6,000 Icelandic individuals might not be generally applicable to other populations.

8. The UKB dataset (ref 49) has been published, but was not used in this study of microsatellite mutation. Reference to this dataset in the results and Figure 1 is misleading and confusing, and should be removed.

9. Discussion, line 224- the term "replication adducts" is incorrect in this context. A DNA adduct is a covalent modification to the DNA structure, such as a base lesion. The authors are referring to slipped/misaligned DNA strands, which are formed transiently during replication/repair/recombination, and are not adducts.

Reviewer #1 (Remarks to the Author):

This manuscript presents an interesting preliminary analysis of microsatellite mutations in a large set of trios. Echoing findings about single nucleotide polymorphisms, the authors find positive correlations of microsatellite mutation rates with paternal and maternal age, and they also find evidence that genetic variation at at least two loci regulates germline microsatellite stability. The existence of these genetic variants, in particular, will help others interpret variation in microsatellite variation within the human population, since such variation has been studied for decades but always under the assumption that mutation rates are constant between genotypes and populations.

Although the paper's results seem interesting and plausible, the paper is hindered by a lack of methodological detail. In some cases, key details are supplied in the supplement, but are omitted from the main text in a misleading or confusing way. In other cases, the underlying analysis may not be rigorous enough for the results to pass muster.

One weakness of the paper with potentially far-reaching downstream consequences is a failure to estimate an accurate false positive rate for homopolymer mutations. The authors note that even HiFi data has a high error rate at homopolymer loci, and short read data certainly has a higher error rate at these loci compared to repeats of motif length greater than 1. The transmission rate of 0.4 at homopolymer mutation sites suggests that at least 20% of homopolymers calls are likely to be false positives. The number of false positives is actually higher than this in all likelihood, since third generation transmission is unable to detect false positives that are caused by allelic dropout, i.e. genotyping errors in the parents that cause variants transmitted to the offspring to falsely look like de novo mutations. Allelic dropout also invalidates the premise of the monozygotic twins analysis, since a genotyping error in one of the parents of a set of twins may cause the same false positive de novo mutation to be called in both twins. To tighten up their validation scheme, the authors need to discuss the possibility of allelic dropout and attempt to minimize it perhaps by requiring that a variant called as a de novo mutation in a child has no supporting reads in either parent.

Response:

We have moved some of the methodological details from the supplement to the main text.

We agree with the reviewer that the error rate for homopolymers is higher than for other types of microsatellites. This error rate is due to the sequencing technology used and is out of our control. We have now made this clear to the reader early in our results section. We do not agree with the reviewer that the high error rate of homopolymers has “potentially far-reaching consequences”. Prior to submitting the manuscript we considered whether the manuscript would be better if results for homopolymers were not presented. Our decision was to present all analyses both with and without homopolymers, hence the reader is made aware of the fact that homopolymers have a higher error rate and is able to gauge both sets of results. These results show that including the homopolymers in the analysis would not lead us to vastly different conclusions. We further note that the high sequencing error rate of homopolymers does not affect our association result since we also performed our association analysis

excluding mDNMs at homopolymer microsatellites. This analysis uncovered the NEIL2 signal which did not appear when the phenotype included mDNMs at all motif lengths and the MSH2 association also remains significant when excluding homopolymers.

We now also mention allelic dropout in the parent as an additional source of error and columns tabulating how many of the DNMs found have a supporting read in the parent have been added to table 1. We note that the presence of reads supporting the mDNM allele in a parent does not mean that there has been allelic dropout as somatic mutations and sequencing errors of microsatellites are quite common, making these variants more difficult to analyze than SNP and indel variants. Further, our previous study of SNPs and indels has shown that de novo variants in a child are frequently present at low allelic frequency in the parent (<https://www.nature.com/articles/s41588-018-0259-9>).

They also need to discuss the results of the validation in the main text (including the percentage of each type of variants that validate by each method) rather than confining the results of these tests to the supplement. Finally, they need to mention much earlier in the manuscript that homopolymer mutations have poor validation rates, disclose how many of their calls are homopolymer mutations, and put error bars around all mutation rate estimates that reflect the high error rate of homopolymer mutations. Could false positive homopolymer mutations be responsible for the apparent enrichment of maternal mutations for homopolymer variants?

Response:

Thank you, we have now added a paragraph discussing in detail the results of each validation method highlighting the poor performance of homopolymers. Table 1 lists the number of mutations by motif length (homopolymers account for 22.3%). Last, we agree that the false positive homopolymer mutations could be responsible for at least some part of the maternal enrichment.

Another analysis that is lacking in methodological detail is the fine mapping component of the GWAS. The authors never explain how they narrow down their first peak to a single SNP in MSH2, or how they narrow down their second peak to a synonymous SNP in NEIL2 (which being synonymous seems unlikely to have a large functional effect). My impression is that the typical GWAS peak is associated with quite a wide confidence interval containing several genes. I assume that the authors picked MSH2 and NEIL2 from within larger confidence intervals because of these genes' prior associations with microsatellite instability, which is reasonable, but the paper needs to be more transparent about how the selection was done, how many other genes are within the peak confidence interval, and whether any of them have functions related to DNA repair or other processes that could impact microsatellite mutation rates.

Response:

The reviewer is correct that narrowing association peaks in GWAS analysis can often be quite difficult, as there are typically a large number of correlated variants in a large LD block containing multiple genes. We used a method presented by Sveinbjörnsson et al. (Nat Gen 2016), cited in the manuscript. Here variants are weighted by their sequence annotation and the variant with the lowest sequence-annotation-weighted p-value is selected as the lead marker. Selecting lead markers for the two loci presented here was also simplified by the fact that the LD blocks were comparatively small, with only three genome-wide significant markers near MSH2 and two near NEIL2. In most cases, lead

markers do not reside within the coding regions of a gene, but in both cases in this manuscript the markers do in fact reside within coding regions. As both lead markers reside within coding regions we believe that the genes in question are the most likely players in the association.

In summary, there is no wide confidence interval to narrow and no other genes with genome-wide significant SNPs. For MSH2, the result is only this SNP and 2 other SNPs (Table S5). For the NEIL2 signal, there are only two genome-wide significant SNPs, both in NEIL2 and the one we report has the lowest p-value and resides within a coding exon. Figure 4 has Manhattan and locus plots that should show this for both cases.

Finally, given that the central point of this paper is to measure a microsatellite mutation rate, I was struck by the fact that the abstract does not actually mention a mutation rate, but only a raw count of de novo mutations per genome per generation. At several points, the paper mentions “high quality” microsatellites and alludes to the fact that the authors only call mutations at a subset of high-quality loci, but this is not discussed at length enough to make it obvious how the raw number of mutations per genome per generation is translated into a mutation rate per locus. A supplementary note also mentions that the sample size of the cohort affects the mutation rate measurement, which seems counterintuitive and is not something that is true of SNV mutation rate measurements. I think it would be more appropriate for the main text to discuss estimation of a mutation rate per locus per generation in a manner that is comparable to other studies, given that this is a central challenge addressed by papers that estimate single nucleotide de novo mutation rates.

Response:

Thank you. As the mutation rate varies considerably across the type of microsatellite and a number of other factors we decided against presenting a single number in the abstract. We instead present the number of mutations inherited by a child, which cannot be easily estimated without having first computed the mutation rate.

We believe that our estimate of mutations per generation is comparable to what has been presented in other studies. DNM rate for microsatellites is generally estimated only at polymorphic microsatellites, while DNM rate for SNPs is typically estimated at all base pairs in the genome (or those that can be reliably characterized). Larger cohorts will necessarily have more polymorphic microsatellites. The additional microsatellites found in large cohorts but not in smaller ones will be enriched for microsatellites that have a low mutation rate, leading to different estimates of the mDNM rate depending on the cohort size.

We have added these points to the manuscript.

Reviewer #2 (Remarks to the Author):

The manuscript by Kristmundsdottir et al. used a novel Icelandic parent-offspring trio WGS dataset to estimate the incidence of de novo microsatellite mutations (mDMNs) in humans. Their analysis was inclusive of 1-6 bases/unit microsatellite classes (e.g., mononucleotide through hexanucleotide repeats). The first part of the study confirms the impact of inherent DNA variables (length, sequence, purity) on microsatellite mutation that has been well established by numerous previous studies of humans (e.g., 1000 genomes), primates, and other organisms. From this foundation, the authors examined the association of mDMNs with parental age and sex to deduce additional factors affecting microsatellite mutation. Finally, the authors performed a GWAS analysis to discover SNPs associated with mDMN incidence.

In general, the methods used are straight-forward and add important knowledge regarding germline rates of microsatellite mutation in humans. Parental biases in microsatellite mutation have been documented for a few microsatellites that cause disease, and this study now provides a genome-wide analysis of parental biases (sex and age) which is an important advancement. However, while the GWAS section is potentially the most innovative and biologically significant section of the study, insufficient evidence is provided to support the stated major conclusion that this study shows the “microsatellite mutation rate in humans is in part under genetic control”. The authors overinterpret and overstate the significance of the GWAS results, and additional data are required to support the conclusion that the identified variants in MSH2 (rs4987188) and NEIL2 (rs8191642) are “microsatellite mutators”. The authors state (lines 211-215; Supplementary note 9) that their results of an association between the variants and mutation are confined to microsatellites, as they found no association with base substitution mutational signatures. If true, this would be an unprecedented result, because no genetic mutation that solely impacts microsatellite mutation without impacting single nucleotide substitutions or indels has been demonstrated in over 2 decades of microsatellite mutation research. The alternative explanation of their results is that the SNP associations have no mechanistic impact on mDNMs. Because of this, it is imperative that the same results be shown by another approach (for instance, an experimental model system).

Response:

We agree with the reviewer that further orthogonal evidence of the association results would be of great interest. We believe that additionally performing experiments in an experimental model system to be outside the scope of this manuscript.

We now discuss the association result in greater detail in the manuscript. The association that we observe is highly significant, even after correcting for the number of markers tested genomewide. An association could be due to 1) carriers of the germline variants having higher mutation rate 2) Individuals with high germline mutation rate being more likely to carry the variant 3) a common source to both the genetic variant and the somatic variants. We find explanation 1) to be the most likely one, but now for absolute clarity we mention in the manuscript that the other explanations are also possible. We are not aware of any biological mechanism that would support 2). Explanation 3) could in theory happen due to sequencing error, however the variants are highly reliably characterized, as measured by transmission in pedigrees and imputation information. The fact that one of the variants is in MSH2, a gene that is known to act in DNA mismatch repair further adds evidence in support of 1).

Additional concerns and suggestions for improvement are provided in detail below.

1. Association of mDMN incidence with DNA repair gene SNPs. This is the weakest section of the study. The MSH2 missense variant has been studied in a yeast model (references 46 and 47). In these publications, it was shown that the variant is functional (able to complement an MSH-deficient cell), and the authors of ref. 47 conclude that any impact on mutation rates would be due to levels of protein expression. In the current study, Kristmundsdottir et al. provide no data showing that expression of the MSH2 variant protein is altered, compared to wild-type. Furthermore, they found no association of the variant allele with cancer risk (as would be expected for a defective MSH2 allele), suggesting the variant is functioning normally in maintaining genome stability. The association with the NEIL2 variant is not compelling. This is a synonymous variant and no literature exists to demonstrate this SNP alters NEIL2 function. Furthermore, no evidence is provided by the authors or cited from the literature to support a role for NEIL2 and base excision repair in microsatellite mutagenesis.

Response:

We believe that the association results that we present are robust. The exact mechanism through which the variants affect mutation rate is something of great interest. This is not something that we have attempted to answer in this manuscript, but we expect that follow up studies will attempt to address these questions.

Despite this, we wish to clarify that the conclusions drawn in ref.47 (ref. 55 in revised version) are not that the impact on mutation rates would be due to the variant's effect on protein expression, rather that the mutated rs4987188 allele would need higher levels of expression to fully complement an *msh2Δ*-null mutant. The statements made regarding the yeast homologue's (G317D) effect on mutation rate are the following:

1.Cells expressing G317D, however, exhibit an MMR defect of 1.7, and this difference from YBT25 complemented with wild-type MSH2 appears significant (P=0.0073). These results indicate that the G317D replacement encodes a functional protein of slightly reduced efficiency in MMR

2.In a previous study (36), the yeast G317D allele partially complemented an msh2Δ-null mutant when expressed at high levels from a GAL10 promoter, but did not provide any complementation when expressed from the native MSH2 promoter. Cumulatively, these results indicate that the G317D allele is an efficiency polymorphism and that in vivo function of this variant may be sensitive to the levels of expression.

(Link to paper: <https://academic.oup.com/hmg/article/10/18/1889/2901493?login=true>)

The key sentence is that G317D is a polymorphism that affects the MMR efficiency and that it is sensitive to the levels of expression. From this it is clear that the authors of ref. 47 conclude that for the mutated rs4987188 allele to be equivalent to the wild-type it needs to be expressed at higher levels, further supporting our results.

To emphasize how this supports our results, we have added a boxplot of the MSH2 rna-expression for each rs4987188 genotype, (0/0),(0/1) and (1/1) and show that they are in fact not different from each other in our data. From this we can conclude that the MMR efficiency in carriers should be decreased relative to non-carriers, and that their mutational load should be increased.

MSH2 expression in blood

The fact that we have not found an association of MSH2 to cancer risk may be due to 1) limited power in our study or 2) a difference in the function of MSH2 in somatic compared to germline tissue.

As for the NEIL2 signal, we agree that the literature is limited with respect to NEIL2's function in microsatellite mutagenesis. It has however been repeatedly shown to function as a part of the BER pathway, more specifically in replication coupled repair and especially within DNA bubble structures which we believe strengthens our conclusion that NEIL2 is the most likely player in the association. We tested the association of rs8191642 to expression of NEIL2 in blood and found that it is correlated with the

genotype in this case (Effect -0.561 SD, p-value 1e-412).

NEIL2 expression in blood

2. The manuscript's methods should be improved to provide more transparency and allow others to reproduce the results. For the RRT, did the authors use a lower limit for the number of repeat units to define a microsatellite (e.g., were 2 unit motifs as in GG included)? Was this lower limit the same for all units (mono, di, tri, etc)? This is important, as published microsatellite analyses use different definitions. At the other end, was there an upper limit to the RRTs? The authors imply (lines 85-88) that genotypes above 80 bp were unreliable, so was 80bp the upper limit for all STRs? Lastly, lines 463-464 state that WGS regions were removed/excluded. The authors should provide the number of bases or % of the whole genome sequence that were actually included in the analyses.

Response:

Thank you, we mention in the text that our upper RRT length limit is 144 bp and we have now added a table with the minimum RRT length for each motif length. Further, we now explicitly state that our high quality marker set contains 45% of the 1,394,292 microsatellites we discover in the Icelandic set which in turn represent 26% of the full set of 5,401,401 STRs we examined, the remaining 4,007,109 STRs were non-polymorphic.

The set of DNMs is made available with the publication. We determined the set of STRs that we tested using the software program Tandem Repeats Finder (TRF) and the set of STRs tested are available on the github page of popSTR (<https://github.com/DecodeGenetics/popSTR>).

3. The authors discuss only one possible mechanism underlying the sex bias observed, that oocytes sustain more DNA damage than sperm. They state “It seems likely”, but provide no scholarly references to support the hypothesis that oocyte damage is greater than sperm damage. The rationale for the hypothesis needs to be more developed, and the paper would be strengthened by citing publications showing that oxidative or alkylation damage increases microsatellite mutation. The alternative hypothesis, that DNA replication or DNA repair programs differ in oocytes and sperm, are equally plausible and should be discussed. Also, parental biases in microsatellite mutation have been well documented for microsatellite expansion diseases. The authors should also cite biases known from the literature (in addition to SCA10). For instance, DM-1 (myotonic dystrophy) shows a paternal bias, while FRAX mutations show a maternal bias.

Response:

The reviewer raises an interesting point. We have now noted that alternate explanations for the sex bias may exist. We have also added discussion on and cited a paper covering the DM1 repeat expansion parental bias. Since chrX is not included in our set we are however unable to examine the maternal bias at FRAX mutations.

4. The results showing the size of maternal versus paternal mDNMs are not clearly presented and it is difficult to independently assess the authors’ conclusions. The authors state in the Abstract (line 22) that “maternal mDMNs affect more bp”, a statement that is ambiguous. What does “more bp” mean, and how was this calculated? Does the “more” mean a comparison to parental mDMNs? Table 3 presents the size of mDMNs from each parental sex, by microsatellite class. However, the conclusion stated in the table legend and in the Results (lines 124-126) that “maternal mDMNs are larger on average than parental ones” is not apparent in the data presented.

Response:

By “more base pairs” we meant to say that maternal mDNMs are on average larger than paternal mDNMs. We have now corrected this in the revised manuscript and clarified the tables suggested.

5. The biological significance of the age effects measured should be discussed. The authors state that mDNMs increase by 0.31/year for maternal age at conception. Given the human window for conception spans 30 years (~15 yoa to 45yoa) on average, this corresponds to an increase, at most of 10 mDMNs due to maternal age, or ~15 % of the overall rate stated (64 mDMNs/offspring/generation).

Response:

In the previous version of the manuscript (page 8), we have stated the number of mutations transmitted by 20 and 40 year old mothers and fathers. Although this is not the full span of the window for conception we believe that most conceptions occur in this time window.

6. Introduction (lines 51-59) states that “apart from a handful of cases, no germline mutators are known”. The authors have failed to discuss the well-known constitutional mismatch repair deficiency, caused by homozygous inheritance of mutations in mismatch repair genes. These references should be included in the manuscript.

Response:

Our statement referred to segregating variants in a population that affect the number of germline mDNMs transmitted from parent to offspring. We have now added a citation to the papers that the reviewer suggests, but we note that those publications refer to mutations that are somatic in the proband. In these cases the mutations accumulate during the life of the proband due to their mismatch repair deficiency. Our variants directly affect the gametes and mutations are present even before fertilization making it unnecessary for the proband to inherit the variants to have an increased number of mutations.

7. The authors should discuss the limitations of this study. Findings from a population of ~6,000 Icelandic individuals might not be generally applicable to other populations.

Response:

We understand the reviewer’s concerns and have added to the discussion a paragraph addressing them. However, humans are a relatively homogenous species and Icelanders are likely to be fairly representative. This has been demonstrated multiple times with GWAS results from Iceland and estimation of the de novo SNP mutation rate using Icelandic trios. Further, comparison of the polymorphism rates and expected heterozygosity of microsatellites between the UKB and Iceland shows that Icelanders are quite similar to individuals in the UK Biobank.

8. The UKB dataset (ref 49) has been published, but was not used in this study of microsatellite mutation. Reference to this dataset in the results and Figure 1 is misleading and confusing, and should be removed.

Response:

We disagree with the reviewer. It is true that the UKB dataset has been published. However no analysis of the polymorphism rate or expected heterozygosity for microsatellites was performed there. These results are presented in our manuscript for the first time.

9. Discussion, line 224- the term “replication adducts” is incorrect in this context. A DNA adduct is a covalent modification to the DNA structure, such as a base lesion. The authors are referring to slipped/misaligned DNA strands, which are formed transiently during replication/repair/recombination, and are not adducts.

Response: Thank you, we have now edited text to correct this error

REVIEWER COMMENTS

Reviewer #1 (Remarks to the Author):

The authors have done an excellent job with revisions. The results are now presented in a more nuanced way that strikes a good balance between novel contributions and new as-yet-unanswered questions.

Reviewer #3 (Remarks to the Author):

To help address the lack of relationship between the MSH2 variant and SNV mutation rates, it would be useful to compare the number of SNV and microsatellite de novo mutations (adjusted for age and sex) transmitted by each parent in this cohort. This group has published previously on the SNV mutation rate in these trios, so the comparison should be easy to do. If there is no correlation between the number of transmitted de novo SNVs and de novo microsatellite alleles, this would lend indirect support to the notion that different causal mechanisms affect mutations in these two types of polymorphisms. A very recent paper (C. Steely et al., 2022, Genome Biology) carried out this comparison on a smaller sample of pedigrees and found no correlation between transmitted de novo SNV and microsatellite alleles. Incidentally, this paper estimated the average number of transmitted de novo microsatellite alleles to be very similar to the number reported in this study, so it would be worth citing (the paper likely appeared after the authors submitted this ms).

In this revision the authors state that an expanded CAG repeat causes myotonic dystrophy type 1 (DM1) and that there is a paternal bias for repeat expansion. This is incorrect. DM1 is caused by a CTG repeat expansion (sometimes this is labeled CTG/CAG to indicate forward and reverse DNA strands) in the 3' UTR of DMPK. The expansions occur much more frequently with maternal, not paternal, transmission. This information is all described accurately in the review paper cited by the authors (Lanni and Pearson, 2019 – see first three sentences of the abstract). This correction actually works in the authors' favor because they found mild evidence for a maternal bias in CTG repeat expansion in their cohort.

In the GWAS analysis, is there any evidence for associations of variants in/near other known mismatch repair genes (e.g., MLH1, MSH6, etc.) and microsatellite mutations? Even if not genome-wide significant, trends in this direction would be interesting and supportive.

The authors may wish to cite another paper that just appeared last month. P. Vijayaraghavan et al. (2023, PloS One) surveyed STR variation and transmission in a multi-ancestry cohort, with results relevant to this manuscript.

Two of the bioRxiv papers in the bibliography have now been published: Sasani et al. (2022, Nature Vol. 605 Issue 7910 Pages 497-502) and Halldorsson et al. (2022, Science) – I'm sure the authors are aware of the latter.

Reviewer #4 (Remarks to the Author):

First, to avoid implicit bias, I read the revised manuscript first, before reading the previous reviewer's comments.

Second, my overall conclusion is that the statistical aspects of the work do NOT meet the reproducibility standard. This issue, "the paper is hindered by a lack of methodological detail", was raised by Reviewer #1 in the previous round of review. But, unfortunately I do not think the revision addressed this major concern.

In fact, I had much trouble understanding the various models and tests used throughout the revised manuscript. To give some specific examples based on the information provided in the Methods;

In the section called Microsatellite attribute regression on mDNM rate, the whole description of the method reads:

"We performed a Poisson regression using the number of available markers per trio as an offset on the full data set and stratified by motif length to examine if the effects of other attributes on the mDNM rate remained consistent across motif lengths."

As the regression is not the standard GWAS GLM, and data at hand are trio-data, additional details are needed, e.g. the analytical unit (offspring vs parents vs all, and if relevant how to deal with related sample), the response variable (and zero inflation concern here), and all the covariates included in the model.

In the section called obtaining confidence intervals for mDNM rate estimates, the whole description of the method reads:

“We used the boot package for R to obtain confidence intervals for both our genome wide mDNM rate estimate and the motif length specific estimates. 100 replicates were used in all cases and 95% confidence intervals extracted using the resulting quantiles.”

Again, there are no sufficient method details to reproduce the research. What was the input data? How were the replicates generated and under what assumptions?

In the section called Parental age effect regression, the authors refer to an earlier work (Jónsson, H. et al. 2017) for the Poisson regression model used. This is generally fine if the model was a standard GWAS type of GLM. However, I note that the model described in Jónsson, H. et al. 2017 was rather complex, with discussions of the effects of phasing, as well as the starting points of the nonlinear optimizing function nlm, on the performance of the model. Additionally, the authors of this manuscript then “compute the total predicted number of de novo mutations in a offspring with an X year old father and a Y year old mother from the coefficients from our regression model”. Thus, it is imperative to describe in sufficient detail (and discuss) the Poisson regression model used.

Third, another major, and immediate, concern I had was how do we ever separate de novo mutations (especially when the rate is in the $e-4$ scale) from technological errors?! Upon reading Reviewer #1’s comments, I learned that this was indeed a major concern. The revision appears to include many new materials addressing this issue, but I do not have the expertise to evaluate the quality of the newly added work.

Reviewer #1 (Remarks to the Author):

The authors have done an excellent job with revisions. The results are now presented in a more nuanced way that strikes a good balance between novel contributions and new as-yet-unanswered questions.

Response: Thank you so much, we are happy to hear this and thank you again for all your helpful comments and suggestions.

Reviewer #3 (Remarks to the Author):

-The issue of whether the two proposed mutator genes affect SNVs as well as STRs has not been addressed. It should be addressed in the main text.

Response: We had compared the number and types of transmitted DNMs in carriers and non-carriers for both variants in the main text and have included a Supplementary note further describing the comparison. We have now computed the correlation between the number of transmitted DNMs and mDNMs and added a supplementary table with these values.

-The argument outlined in the response to Reviewer #2, regarding yeast studies and expression data as support for variant functionality, should be included in the supplementary text.

Response: This has now been added as Supplementary note 9.

-The potential association between the NEIL2 variant and microsatellite instability should be described as being much more speculative than that of MSH2

Response: We have made note of this in the manuscript. We note that following the recommendation of the reviewer we performed an enrichment analysis for associations of variants in known mismatch genes, where we do observe a nominal enrichment of associations in NEIL2, further supporting its role in microsatellite mutagenesis.

To help address the lack of relationship between the MSH2 variant and SNV mutation rates, it would be useful to compare the number of SNV and microsatellite de novo mutations (adjusted for age and sex) transmitted by each parent in this cohort. This group has published previously on the SNV mutation rate in these trios, so the comparison should be easy to do. If there is no correlation between the number of transmitted de novo SNVs and de novo microsatellite alleles, this would lend indirect support to the notion that different causal mechanisms affect mutations in these two types of polymorphisms. A very recent paper (C. Steely et al., 2022, Genome Biology) carried out this comparison on a smaller sample of pedigrees and found no correlation between transmitted de novo SNV and microsatellite alleles. Incidentally, this paper estimated the average number of transmitted de novo microsatellite alleles to be very similar to the number reported in this study, so it would be worth citing (the paper likely appeared after the authors submitted this ms).

Response: Thank you for an excellent suggestion, we have added a citation to the paper suggested and estimated R^2 between maternally and paternally transmitted sDNMs and mDNMs and added a table with these values. We see very modest correlations ($r^2 < 0.04$) between sDNMs and mDNMs.

In this revision the authors state that an expanded CAG repeat causes myotonic dystrophy type 1 (DM1) and that there is a paternal bias for repeat expansion. This is incorrect. DM1 is caused by a CTG repeat expansion (sometimes this is labeled CTG/CAG to indicate forward and reverse DNA strands) in the 3' UTR of DMPK. The expansions occur much more frequently with maternal, not paternal, transmission. This information is all described accurately in the review paper cited by the authors (Lanni and Pearson, 2019 – see first three sentences of the abstract). This correction actually works in the authors' favor because they found mild evidence for a maternal bias in CTG repeat expansion in their cohort.

Response: Thank you, this error has now been corrected.

In the GWAS analysis, is there any evidence for associations of variants in/near other known mismatch repair genes (e.g., MLH1, MSH6, etc.) and microsatellite mutations? Even if not genome-wide significant, trends in this direction would be interesting and supportive.

Response: This is an interesting point, we looked for suggestive associations in genes participating in mismatch repair, base-pair excision repair and nucleotide excision repair, found using the gene ontology resource (<https://www.ebi.ac.uk/QuickGO/>)

For each gene, we tested for association enrichment in six different phenotypes quantifying the number of transmitted mDNMs: all parents, fathers, mothers and the same three excluding homopolymers. We observe nominal enrichment of associations for all six phenotypes in NEIL2 and nominal associations in one or more phenotypes for a number of other genes, including MSH3 (four out of six) and MLH1. These results are now discussed in the manuscript and we have added a supplementary table with p-values and association counts in all the genes we tested.

We have also searched specifically for associations in coding variants in these genes and when correcting only for variants in these genes we find two additional variants in MSH3 that associate with our phenotypes.

All our association results will be published on the deCODE genetics website (decode.com/summarydata).

The authors may wish to cite another paper that just appeared last month. P. Vijayaraghavan et al. (2023, PloS One) surveyed STR variation and transmission in a multi-ancestry cohort, with results relevant to this manuscript.

Response: Thank you, we have added a reference to this paper.

Two of the bioRxiv papers in the bibliography have now been published: Sasani et al. (2022, Nature Vol. 605 Issue 7910 Pages 497-502) and Halldorsson et al. (2022, Science) – I'm sure the authors are aware of the latter.

Response: Thank you, these references have now been updated.

Reviewer #4 (Remarks to the Author):

First, to avoid implicit bias, I read the revised manuscript first, before reading the previous reviewer's comments.

Second, my overall conclusion is that the statistical aspects of the work do NOT meet the reproducibility standard. This issue, "the paper is hindered by a lack of methodological detail", was raised by Reviewer #1 in the previous round of review. But, unfortunately I do not think the revision addressed this major concern.

In fact, I had much trouble understanding the various models and tests used throughout the revised manuscript. To give some specific examples based on the information provided in the Methods;

In the section called Microsatellite attribute regression on mDNM rate, the whole description of the method reads:

"We performed a Poisson regression using the number of available markers per trio as an offset on the full data set and stratified by motif length to examine if the effects of other attributes on the mDNM rate remained consistent across motif lengths."

As the regression is not the standard GWAS GLM, and data at hand are trio-data, additional details are needed, e.g. the analytical unit (offspring vs parents vs all, and if relevant how to deal with related sample), the response variable (and zero inflation concern here), and all the covariates included in the model.

Response: We have now extended this section to further describe the regression formula used and what parameters were passed. We have also stated explicitly what calls were made to the software used and committed the source code and the input we ran it on to a public github repository.

In the section called obtaining confidence intervals for mDNM rate estimates, the whole description of the method reads:

"We used the boot package for R to obtain confidence intervals for both our genome wide mDNM rate estimate and the motif length specific estimates. 100 replicates were used in all cases and 95% confidence intervals extracted using the resulting quantiles."

Again, there are no sufficient method details to reproduce the research. What was the input data? How were the replicates generated and under what assumptions?

Response: Thank you, to address this clear lack of details we have added a description of the input, clarified how the R package we used generates replicates and added the script along with its input to a public github repository.

In the section called Parental age effect regression, the authors refer to an earlier work (Jónsson, H. et al. 2017) for the Poisson regression model used. This is generally fine if the model was a standard GWAS type of GLM. However, I note that the model described in Jónsson, H. et al. 2017 was rather complex, with discussions of the effects of phasing, as well as the starting points of the nonlinear optimizing function nlm, on the performance of the model. Additionally, the authors of this manuscript then “compute the total predicted number of de novo mutations in a offspring with an X year old father and a Y year old mother from the coefficients from our regression model”. Thus, it is imperative to describe in sufficient detail (and discuss) the Poisson regression model used.

Response: This is a good and important point, to adequately address it we have recapitulated the model described by Jónsson et al. We have added further details of our use of the model, explaining how we integrated out the number of available markers for each trio before fitting the model. Further, we have added both the model’s R source code and the input we ran it on to a public github repository. We hope this will improve replicability and clarify our process.

The github repository (https://github.com/DecodeGenetics/mDNM_analysisAndData/) contains the R code used for the analyses mentioned above. The repository also contains the files used as input to the analyses.

Third, another major, and immediate, concern I had was how do we ever separate de novo mutations (especially when the rate is in the $e-4$ scale) from technological errors?! Upon reading Reviewer #1’s comments, I learned that this was indeed a major concern. The revision appears to include many new materials addressing this issue, but I do not have the expertise to evaluate the quality of the newly added work.

Response: We agree that the separation of mDNMs and technological errors is indeed very important. We have taken numerous steps, both before initial submission and added other analysis after revision to address this. As you mention, these were concerns expressed by reviewer one in the previous round of revision but however, after performing modifications and extensions suggested by him, his reply (shown below) leads us to believe that they have been adequately addressed.

Reviewer 1 response in round 2 of revision: *“The authors have done an excellent job with revisions. The results are now presented in a more nuanced way that strikes a good balance between novel contributions and new as-yet-unanswered questions.”*

REVIEWERS' COMMENTS

Reviewer #3 (Remarks to the Author):

I have only two relatively minor comments:

For the correlation between the number of DNMs and mDNMs, the authors state in the text that it is “low” with a reference to Table S8 for the actual values. Please put the two R2 and p values in the main text so readers don’t have to sift through supplementary tables to find it.

I appreciate that the authors followed the suggestion to look for associations between microsatellite mutations and known mismatch repair genes, but I found this sentence unclear: “After correcting for the number of genes tested, we observe a nominal enrichment of associations for all six phenotypes in NEIL2 and for one or more in several other genes (Table S9).” I think they mean “for one or more of the phenotypes in several other genes...”

Reviewer #4 (Remarks to the Author):

The authors have sufficiently addressed my questions. Thank you.

REVIEWERS' COMMENTS

Reviewer #3 (Remarks to the Author):

I have only two relatively minor comments:

For the correlation between the number of DNMs and mDNMs, the authors state in the text that it is “low” with a reference to Table S8 for the actual values. Please put the two R2 and p values in the main text so readers don’t have to sift through supplementary tables to find it.

Response: Thank you, we have added the values requested to the main text.

I appreciate that the authors followed the suggestion to look for associations between microsatellite mutations and known mismatch repair genes, but I found this sentence unclear: “After correcting for the number of genes tested, we observe a nominal enrichment of associations for all six phenotypes in NEIL2 and for one or more in several other genes (Table S9).” I think they mean “for one or more of the phenotypes in several other genes...”

Response: We agree the sentence was unclear as it was and have updated it in the way you suggested to clarify the intended meaning, thank you.

Reviewer #4 (Remarks to the Author):

The authors have sufficiently addressed my questions. Thank you.

Response: We are happy to hear our edits and additions were adequate.